# N-Linked Glycosylation Modulates Golgi-Independent Vacuolar Sorting Mediated by the Plant Specific Insert

**DOI:** 10.3390/plants8090312

**Published:** 2019-08-30

**Authors:** Vanessa Vieira, Bruno Peixoto, Mónica Costa, Susana Pereira, José Pissarra, Cláudia Pereira

**Affiliations:** 1Faculdade de Ciências da Universidade do Porto, Rua do Campo Alegre, s/nº, 4169-007 Porto, Portugal; 2Instituto Gulbenkian de Ciência, Rua da Quinta Grande 6, 2780-156 Oeiras, Portugal; 3GreenUPorto-Sustainable Agrifood Production Research Center, Campus de Vairão, Rua Padre Armando Quintas 7, 4485-661 Vila do Conde, Portugal

**Keywords:** Plant Specific Insert, Aspartic proteinase, vacuolar sorting, unconventional trafficking, endoplasmic reticulum, Golgi, N-linked glycosylation

## Abstract

In plant cells, the conventional route to the vacuole involves the endoplasmic reticulum, the Golgi and the prevacuolar compartment. However, over the years, unconventional sorting to the vacuole, bypassing the Golgi, has been described, which is the case of the Plant-Specific Insert (PSI) of the aspartic proteinase cardosin A. Interestingly, this Golgi-bypass ability is not a characteristic shared by all PSIs, since two related PSIs showed to have different sensitivity to ER-to-Golgi blockage. Given the high sequence similarity between the PSI domains, we sought to depict the differences in terms of post-translational modifications. In fact, one feature that draws our attention is that one is N-glycosylated and the other one is not. Using site-directed mutagenesis to obtain mutated versions of the two PSIs, with and without the glycosylation motif, we observed that altering the glycosylation pattern interferes with the trafficking of the protein as the non-glycosylated PSI-B, unlike its native glycosylated form, is able to bypass ER-to-Golgi blockage and accumulate in the vacuole. This is also true when the PSI domain is analyzed in the context of the full-length cardosin. Regardless of opening exciting research gaps, the results obtained so far need a more comprehensive study of the mechanisms behind this unconventional direct sorting to the vacuole.

## 1. Introduction

In the last years, the outburst of data on molecular mechanisms involved in endocytic and exocytic trafficking have outlined subtle balances between these pathways. The study of specific cases shows that membrane and cargo molecules’ exchange within the cells or in response to the external cell environment is finely tuned [1,2,3]. In higher plants, the presence of two types of vacuoles, sometimes co-existing, necessarily implies the existence of different sorting mechanisms for each one of these organelles [4,5,6]. The recognition of these vacuolar sorting determinants (VSDs) generally occurs at the late or post-Golgi level [5,6], redirecting the protein away from the secretory pathway, and towards the vacuole. However, there are reports showing that the delivery of some vacuolar proteins can be promoted very early in the sorting process, when polypeptides are still contained within the ER [7]. Parallel to the conventional Endoplasmic Reticulum (ER) > Golgi > prevacuolar compartment (PVC) route to the vacuole, a more unconventional sorting to the vacuole that bypasses the Golgi has been described in recent years [8,9,10,11,12]. Often, the relevance of this sorting mechanism depends on the type of tissue/cell they are expressed in and the metabolic activity of such organs.

Cardosins are well-characterized aspartic proteinases extracted from the vascular plant *Cynara cardunculus* [13,14]. Two different, yet related, proteinases were isolated, cardosins A and B, and they have been extensively studied over the years both in the native and heterologous systems [9,15,16,17,18,19]. Each of these enzymes is synthesized as a precursor and undergoes different cleavages along the endomembrane system in order to acquire its mature form, composed of heavy and light chains (Figure 1A).

One domain that is cleaved out during this process is an insert of 100 amino acids termed the Plant-Specific Insert (PSI, Figure 1B). The relevance of this domain for the aspartic proteinase function has been widely discussed mostly because it is only present in some aspartic proteinases [20,21]. Several roles have been attributed to this particular domain namely given its ability to interact with lipid membranes and its putative antimicrobial activity. A number of studies depict these possibilities using PSIs isolated from different plants, and it has been proven that this domain can modulate the behavior of model membranes [22] and is also able to induce membrane permeabilization, leading to the release of vesicle contents [23]. This ability is closely related to its antimicrobial activity: a study with *Solanum tuberosum* aspartic proteinase PSI showed that it enhances *Arabidopsis thaliana* resistance against *Botrytis cinerea* infection [24], and another report using the same PSI showed that it is cytotoxic against Gram-negative and Gram-positive bacteria [25]. All these particular features have raised the interest in this domain and several studies were conducted using PSIs isolated from different plant species in an attempt to better characterize this domain, considered to be “an enzyme inside an enzyme” [24,25,26,27,28].

Though its functions *in planta* are still highly debated, some reports indicated that this domain was responsible for AP vacuolar targeting [9,23,28,29]. In fact, Pereira and co-workers (2013) showed that the PSIs from *C. cardunculus* cardosins A and B are able to redirect secreted proteins to the vacuole and that different PSI domains may follow different routes to the lytic vacuole of *Nicotiana tabacum* leaf epidermal cells, by a process that is not yet clearly understood [9]. A poly-sorting mechanism for cardosin A has been described, with two different vacuolar signals: the C-terminal peptide, a Ct-VSD by definition, and the PSI, a more unconventional sorting determinant. It was demonstrated that each domain determines a different route to the vacuole in *N. tabacum* leaves: The PSI is able to bypass the Golgi, while the C-terminal peptide follows a classic ER-Golgi-PVC route to the vacuole. This poly-sorting mechanism seems to be related to the different roles of the protein *in planta* and associated with specific cell needs [9,15]. A working model for cardosins’ trafficking suggests that the PSI mediates either a COPII-independent or COPII-dependent pathway depending on its glycosylation status, as cardosin A PSI, contrary to cardosin B, is not glycosylated.

To understand both the roles and significance of these two PSI domains in vacuolar targeting, we cloned the PSIs and generated glycosylation mutants for analysis by transient expression in *Nicotiana tabacum* leaf epidermis. Furthermore, we wanted to test if the differences observed with the isolated PSIs would persist when they are expressed in the full-length cardosin. To achieve that, we produced different constructs regarding the full-length cardosin A and B: two with the native PSI domain and two with swapped PSI domains. The results obtained indicate that cardosin A PSI’s ability to bypass the Golgi is maintained in the context of the full-length protein and also that glycosylation seems to have a preponderant role in this process. As a proof of concept, we finally isolated two PSIs from *Glycine max* aspartic proteinases, showing the same glycosylation dichotomy, and the results obtained are in agreement with the role of glycosylation we propose. Overall, we expect to have a better definition on how the PSI ER-to-Vacuole direct sorting is orchestrated, through which mechanism this is occurring, and to define the intermediate players in the process.

## 2. Results

A previous report showed shown that cardosin A’s and cardosin B’s PSI–PSI-A and PSI-B, respectively, are able to sort proteins to the vacuole [9], a characteristic that is believed to be shared by the PSIs of other plant aspartic proteinases. Here we sought to clarify some aspects of their trafficking, focusing on the differences and similarities between them.

### 2.1. PSI Domains Maintain Their Sorting Capacity, Independently of the Overall AP Structure

Different PSI domains are linked with different vacuolar sorting routes in the plant cell and this feature is probably related to their physiological roles (or the role of the aspartic proteinase they belong to) in the native plant, raising the question whether they can retain their sorting properties if in the context of the entire protein. To assess that, we tested the expression and sorting of cardosin A and cardosin B, the chimaeras cardosin A with cardosin B PSI (CardosinA::PSI-B), and cardosin B with cardosin A PSI (Cardosin B::PSI-A), with and without their C-terminal domain (Appendix A).

Knowing that the carboxy termini-mediated vacuolar trafficking is dominant over PSI-mediated trafficking in cardosin A [9], and since we were primarily interested in observing PSI-mediated trafficking dynamics, we began this work by removing the ctVSD from all tested chimaeras (Figure 2A). Upon removal of the C-terminal peptides, the vacuolar accumulation pattern remained observable for all tested fluorescent fusion proteins, indicating that both PSI domains are sufficient for directing either of these aspartic proteases to the lytic vacuole, further confirming the exchangeability of these domains between aspartic proteinase molecules (Figure 2B).

In order to further dissect the specific pathways each of these constructs followed towards the vacuole, we expressed each one of them in cells undergoing blockage at specific points of the secretory pathway. Cardosin A behavior has already been thoroughly explored at the ER-Golgi level, with a clear difference between cardosin A and its truncated version (Cardosin AΔC-ter) being apparent. In particular, the removal of its carboxyl terminus results in an acquired insensitivity towards SarI^H74L^ and RabD2a^N121I^ blockage of COPII vesicle formation, a phenomenon that has been attributed to the route mediated by this PSI domain [9]. On the other hand, removal of cardosin B C-terminus (Cardosin BΔC-ter) did not result in a Golgi-bypass capability, with the fluorescence becoming accumulated within the ER network upon co-expression with the dominant negative mutant form of SarI (Figure 2C d–f). A similar behavior could be observed with the Cardosin A::PSI-BΔC-ter fluorescent protein (Figure 2C g–i).

Co-expression of this protein with the dominant negative mutant SarI^H74L^, or the small GTPase RabD2a^N121I^ (not shown), resulted in an ER accumulation pattern (Figure 2C g–i), suggesting that this protein must pass through the Golgi before reaching the vacuole. These observations allow us to conclude that (i) the PSI-B domain retains its functionality as a VSD in the heterologous proteins tested, and that (ii) it directs aspartic proteases to the vacuole through a Golgi-dependent pathway, whether it is expressed alone or in the context of an AP. Contrary to what could be observed with the truncated cardosin BΔCter, Cardosin B::PSI-AΔC-ter demonstrated the same insensitivity towards the ER-Golgi blockage as could be observed with all other reporters containing the PSI A determinant, accumulating in the vacuole even when co-expressed with either SarI^H74L^ (Figure 2C j–l) or RabD2a^N121I^ (not shown). This result confirms that exchanging the PSI domains between both aspartic proteases results in a shift in terms of the vacuolar sorting route followed by the reporter protein.

### 2.2. Different PSIs Mediate Different Sorting Routes to the Vacuole

Our lab has previously shown that cardosin A has two different vacuolar sorting domains: the PSI and the C-terminal peptide [8]. Upon isolation of the two domains, it was also disclosed that they follow different routes to the vacuole, with the PSI taking a shortcut and driving a Golgi-independent pathway. The question was raised whether all the PSIs have this ability. Cardosin B is another well-studied aspartic proteinase from cardoon, with a similar structure and high similarity in terms of protein sequence [17], in particular regarding the PSI domains (Figure 3A). We cloned the PSI from cardosin B between the signal peptide (SP) from *A. thaliana* chitinase and mCherry–SP::PSI-B::mCh and co-expressed both SP::PSI-A::mCh and SP::PSI-B::mCh (Figure 3B) in *N. tabacum* leaf epidermis with the ER marker GFP::HDEL [30] and the Golgi marker ST::GFP [31]. It is clear that 3 days post-infiltration (dpi), SP::PSI-A::mCh (Figure 3C a–f) and SP::PSI-B::mCh fusions (Figure 3C g–l) are accumulated in the vacuole and there is no fluorescence overlap of the fluorescent proteins with ER and Golgi markers. Both PSI-A and PSI-B are sufficient and efficient in directing mCherry to the vacuole. Next, we sought to depict their route to the vacuole using the dominant negative form of SarI (SarI^H74L^::YFP) [32,33] to specifically block the transport between the ER and vacuole (Appendix A). It has already been documented that PSI-A is able to bypass this blockage and accumulate in the vacuole [9]. Here we show that PSI-B::mCherry is more affected if this pathway is blocked as it gets retained in ER-Golgi compartments (Appendix A), despite some proteins still being able to accumulate in the vacuole. As a control for the action of the dominant negative SarI mutant, SP::mCh::CterA fusion protein was used [9]. This fusion protein travels to the vacuole through the Golgi and prevacuolar compartment, becoming retained if the ER-to-Golgi pathway is blocked (Appendix A).

Additionally, we did a parallel assay using Brefeldin A (BFA) [17,34] to block the ER-to-Golgi transport (Figure 3D). BFA is known to inhibit the function of COPI, resulting in impaired ER export and disruption of the ERES (ER-export sites), thus inhibiting vesicle budding or fusion [35]. As a control, SP::mCh::CterA was again used, and it is possible to observe the absence of fluorescence in the vacuole, with the protein accumulating in the ER and in large aggregates (most probably BFA compartments [34]) (Figure 3D c–d). The results obtained for both PSIs show accumulation of the fusion proteins in aggregates similar to the ones described for SP::mCh::CterA, despite some mCherry fluorescence also being detected in the vacuole for SP::PSI-A::mCh (Figure 3D g–h) and some cells expressing SP::PSI-B::mCh (Figure 3D k–l). Mock treatment with DMSO does not affect vacuolar accumulation of any of the tested constructs (Figure 2D a–b, e–f, i–j). To corroborate these results, the three fusion proteins were also co-expressed with the dominant negative version of RabD2a–RabD2a^N121I^ [36] (Appendix A), leading to the same observation that the SP::PSIA::mCh construct is able to overcome the ER-to-Golgi blockage and accumulate in the vacuole (Appendix A). The interpretation of these results led us to consider that BFA’s effect in PSI-mediated trafficking is more pronounced, probably because BFA affects the PSI export from the ER, by a still unknown mechanism. We cannot discard the possibility that the ER morphology is considerably affected by this assay, which could be the reason for the observed results. In fact, the action of the SarI dominant negative mutant is more specific, and the ER morphology is much less affected than with BFA [37]. To help clarify this aspect, we did a similar BFA experiment, but using protoplasts isolated from infiltrated *N. tabacum leaves*. Similar to the observations made in leaves, all three fusion proteins are retained in the ER and in large aggregates in the cytoplasm (Appendix A). However, a closer look allows to see that SP::PSIA::mCh is also detected in the vacuole (Appendix A), whereas SP::mCh::CterA (Appendix A) and SP::PSIB::mCh (Appendix A) do so to a much lower degree.

### 2.3. N-linked Glycosylation Modulates PSI-Mediated Sorting

The results obtained indicate that, despite the high homology between these two domains, there must exist a specific characteristic driving the Golgi-independent trafficking of PSI-A. One feature that immediately stands out is the presence of a glycosylation motif in PSI-B (NetNGlyc 1.0 Server prediction [38]—Figure 4A, red box) that is absent in PSI-A. To investigate in more detail the effects of glycosylation in PSI-mediated trafficking, we decided to invert the glycosylation status of the PSIs by introducing a glycosylation motif in PSI-A (SETE>NETE–PSI-A^S86N^) and removing the same motif in PSI-B (NETE>SETE–PSI-B^N82S^) by site-directed mutagenesis (Figure 4A,B). Western blot analysis using a monoclonal anti-mCherry antibody (Figure 4C) of the glycosylated and non-glycosylated PSIs shows differences in the proteins’ migration patterns since PSI-B does not migrate as far as PSI-A in the gel. As PSI-A and B’s molecular weights are predicted to be roughly the same, the observed difference must be related with the existence of post-translational modifications in PSI-B. In fact, the non-glycosylated form of PSI-B migrates further in the gel than the native one (Figure 4C), indicating that the presence of the glycan could be, at least in part, responsible for this difference in gel migration. Accordingly, the glycosylated version of PSI-A does present a band with a higher molecular weight than the native PSI-A, indicating the presence of a glycan. Next, we performed an EndoH assay using the native and mutated PSI forms. EndoH is an enzyme that removes the high-mannose glycans that are added in the ER but not the complex glycans generated in the Golgi. The output in a Western blot would be a shift in molecular weight of proteins not transported through the Golgi upon digestion with EndoH. As expected, no shift is visible in the blot (Figure 4D) for either PSI-A or PSI-B mutated version, since they are not predicted to be glycosylated. In contrast, for the PSI-A glycosylation mutant, a decrease in apparent molecular weight between the untreated and treated samples is quite evident (Figure 4D), indicating the presence of high-mannose glycans.

The EndoH sensitive nature of PSI-A^S86N^ glycan points to one of three possibilities: The first one is that the protein does not travel through the Golgi, thus the glycans cannot be modified into the complex type; the second one considers that the native conformation acquired by the protein impairs the access of the glycan to the Golgi-modifying enzymes; and the third one predicts that the protein leaves the Golgi in an earlier step, before the enzymes have time to modify the glycan. Further studies on localization are needed to clarify this situation. Native PSI-B, however, is likely to possess complex type glycans, modified in the Golgi, as it does not seem to be affected by EndoH treatment as no shift can be observed in the blot. It is worth pointing out, that even the non-glycosylated form of PSI-B has an apparent higher molecular weight than both the native and mutated version of PSI-A, an indication that other modifications, rather than N-glycosylation, might exist in this domain.

Next, we analyzed the expression of PSI-A^S86N^ and PSI-B^N82S^ mutants by confocal microscopy together with ER and Golgi markers. Three days after tobacco cells’ transformation, SP::PSI-A^S86N^::mCh and non-glycosylated SP::PSI-B^N82S^::mCh fusions were mostly observed in the vacuole (Figure 4E a–d and g–j, respectively). However, a closer look shows some protein still in the ER possibly still in transit to the vacuole, in particular for SP::PSI-B^N82S^::mCh (Figure 4E g–i). In the same way as what was done for the native form of PSIs, SarI^H74L^ dominant negative mutant was used to block the ER-to-Golgi trafficking (Figure 4F). When co-expressed with the dominant-negative SarI mutant, SP::PSI-A^S86N^::mCh no longer reached the vacuole, becoming retained in the early secretory compartments (Figure 4F a–f). De-glycosylated PSI-B, however, was able to reach the vacuole but not as efficiently as the native non-glycosylated PSI-A as some protein is still retained in the ER when co-expressed with the mutated form of SarI (Figure 4F g–l). Given the novelty of the results obtained and to get a better view of the microscopic observations, quantitative analysis was performed on the results from the observation of more than 90 cells from three independent experiments. For the quantification, we considered that the total number of cells with fluorescent signals defines a 100% value, and among this population, the different localization patterns were then scored. Data obtained from the quantification of fluorescence patterns reflects the subcellular localization observed in confocal microscopy analysis (Figure 4G).

### 2.4. Post-Golgi Trafficking of PSI-B Might Be Modulated by Protein Structure

After having determined cardosin B PSI functionality as a VSD in both cardosins A and B sequences, we aimed at further dissecting its dynamics as an isolated domain, at the post-Golgi level, as previously reported for SP::PSI-A::mCh domain [9], using the fluorescent proteins described in Figure 5A. To this end, we co-expressed SP::PSI-B::mCherry with the dominant negative version of the small GTPase RabF2b-RabF2b^S24N^, which has been previously described as being capable of impairing protein sorting between the Golgi and the PVC [29]. The results were compared to those obtained for PSI-A::mCherry, which is known to bypass the Golgi in its sorting towards the lytic vacuole and thus is not affected by post-Golgi blockage events (Figure 5B c–d). As a positive control, we used the previously described RabF2b-sensitive cardosin A C-terminal peptide [9]–SP::mCh::CterA, which was infiltrated in a separate region of the same leaf as the experimental constructs. As expected, this fusion protein is secreted to the apoplast when co-expressed with RabF2b^S24N^ (Figure 5B a–b). We observed accumulation of the PSI-B::mCherry fluorescence at the periphery of the cell when co-expressed with the dominant negative form of RabF2b (Figure 5B e–f), suggesting a PVC-mediated pathway.

We proceeded with the dissection of the vacuolar sorting route followed by cardosins under direct control of the PSI B domain. Under these experimental conditions, the truncated reporters Cardosin AΔCter and Cardosin B::PSI-AΔCter accumulated in the vacuole when co-expressed with RabF2b^S24N^ (Figure 5C a–b and g–h, respectively), which was expected given the PSI-mediated Golgi-bypass route suggested for these chimaeras. More unexpected were the results obtained for PSI B-mediated vacuolar sorting: When trafficking was blocked at the post-Golgi level, PSI B-mediated vacuolar sorting remained unchanged, as can be observed by permanent vacuolar accumulation of the Cardosin BΔC-ter and Cardosin A::PSI-BΔC-ter chimaeras (Figure 5C c–d and e–f, respectively). These observations contrast with the ones obtained for SP::PSI-B::mCh, that is not able to overcome this blockage.

### 2.5. Proof of Concept: the Soybean PSIs Case

At least five aspartic proteinases have been identified in the soybean plant [*Glycine max* (L.) Merr.] SoyAP1 and soyAP2 were chosen as two representative soybean aspartic proteinases, due to the high similarities between their amino acid sequences and localization. Both have been isolated and characterized [39]. Like cardosins, soyAPs accumulate in different organs in the plant and show different localization and expression patterns. Furthermore, both have a PSI domain, and it has been suggested by Terauchi and co-workers (2006) that soyAP2-PSI is relevant for protein sorting, but not soyAP1-PSI [28]. Given our work on cardosin PSIs, we became interested in these proteins given their potential as vacuolar sorting determinants and because they share the same glycosylation pattern as cardosins’ PSIs: PSI-1 is glycosylated, while PSI-2 is not (Figure 6A–red box). We isolated the two Soybean PSIs and cloned them in frame with the chitinase signal peptide and mCherry (Figure 6B), in a similar manner to how the cardosin PSIs were designed. The first step to analyze and compare the behavior of the soy PSIs was to do a Western blot and Endo-H digestion experiment, to see if they behaved as the cardosins’ PSI domains. Two characteristics immediately stood out in the blot: The PSI-2 was expressed at lower levels than PSI-1 (equal loadings were used) and there were no migration differences between the two PSIs, as was observed for PSI-B (Figure 6C). Regarding glycosylation status, for PSI1, a shift could be observed upon treatment with Endo H, consistent with the presence of high-mannose glycans. The result obtained contrasts with the one observed for PSI-B, whose glycans are of the complex type. In the case of PSI-1, either the glycosylation sites are not accessible to Golgi-modifying enzymes, or the protein leaves the Golgi in a very early step before Golgi enzymes are able to modify the glycans, or the protein bypasses the Golgi.

Next, we proceeded to evaluate the sorting ability and trafficking pathway of SP::PSI-1::mCh and SP::PSI-2::mCh. After 3 days of transient expression in tobacco leaves, both SP::PSI-1::mCh and SP::PSI-2::mCh accumulated in the vacuole (Figure 6D a–b and c–d, respectively). In some cells, mCherry fluorescence is visible, accumulating at the cell periphery, but this is probably an artefact of overexpression, since increasing or decreasing the OD of the infiltrated fluorescent protein resulted in more or less cells, respectively, with fluorescence in the apoplast (not shown). Co-expression with SarI^H74L^, which impairs trafficking between the ER and Golgi, revealed differences between the two PSIs, as SP::PSI-1::mCh (glycosylated) is not able to reach the vacuole, becoming retained in the early endosomal compartments (Figure 6E a–c), while SP::PSI-2::mCh is not affected by this blockage (Figure 6E d–f). Moreover, treatment with BFA resulted in the accumulation of both constructs in BFA body-like structures (Figure 6F), regardless of some protein still being detected in the vacuole in the case of SP::PSI-2::mCh (Figure 6F c–d). In order to confirm these observations, blocking of the same pathway using co-expression with RabD2a^N121I^ was also used. Data obtained was in accordance with the results observed for BFA and SarI dominant negative mutant, as SP::PSI-2::mCh is not affected by the blockage (Appendix A), while SP::PSI-1::mCh is retained in the ER (Appendix A). The results obtained with the soyPSIs are quite similar to the ones observed for cardosins’ PSIs, in terms of accumulation and glycosylation effect, giving us extra confidence in the considered hypothesis.

## 3. Discussion

This report focuses on another ability of the PSI, that is the sorting of proteins to the vacuole, a characteristic that could in fact be related to the functions mentioned above. Therefore, understanding the mechanisms of PSI-mediated sorting and trafficking inside the cell might as well help elucidate some of its other functions.

### 3.1. A Role for PSIs in Vacuolar Sorting

The majority of aspartic proteinases containing a PSI domain accumulate in the vacuole, and the role of the PSI in vacuolar sorting has long been discussed and was effectively tested in different studies using cardosins and soybean aspartic proteinases [9,28,29]. The study of the vacuolar sorting capability of the PSI in the case of cardosins is challenged by the presence of the C-terminal VSD also present in the precursor form of the enzyme. In the report by Pereira et al. (2013), it was shown that as long as the typical C-terminal VSD is carried by the protein, the PSI domain is not necessary for the protein to reach the plant vacuole. However, in the absence of C-ter VSDs, the PSI domain acts as a true VSD, being sufficient and necessary for correct vacuolar targeting of aspartic proteinases [9]. This is true for the two PSIs studied—cardosin A’s PSI and cardosin B’s PSI—but a major difference was observed by our team and further explored in this study using SarI^H74L^ coupled to a YFP, allowing the direct visualization of the cells co-expressing the two proteins. While PSI-B uses a COPII-dependent pathway from the ER to the Golgi, the PSI-A pathway to the vacuole is independent of COPII carriers, thus not being affected by the blockage provided by SarI^H74L^. Comparing the effect of co-expressing SarI^H74L^ with SP::mCh::CterA or with SP::PSI-B::mCherry fusion, the results are significantly different as some PSI-B::mCherry fluorescence in the vacuole is still visible, while all the C-terminal fusion protein is retained in early compartments. It is fair to say that some of the PSI-B::mCherry protein is independent of COPII transport or that the effect of SarI^H74L^ only delays its accumulation in the vacuole. Nevertheless, the differential accumulation of PSI-A and PSI-B when co-expressed with the dominant negative form of SarI is clear, PSI-A mCherry fusion accumulates in the vacuole not being affected by the blockage, while PSI-B mCherry fusion is retained in the ER. The question raised at this point was: what is the difference between the two PSI domains underlying this differential behavior? The two PSIs are quite similar in terms of protein sequence, and the one feature that stands out upon analysis is the presence of an N-linked glycosylation site in PSI-B. We therefore hypothesized that post-translational modification could be a key component in determining the route to be taken by the PSI-driven targeting. In fact, studies on phytepsin [29] revealed that the transport of this AP is COPII-mediated and that the PSI domain was essential for phytepsin’s transport through the Golgi. Interestingly, as for most common APs (including cardosin B), phytepsin has a conserved glycosylation site in the PSI domain.

### 3.2. A Putative Role for Glycosylation in Sorting?

The role of glycosylation in the vacuolar routes taken by proteins in the secretory pathway has long been discussed, in particular involving Golgi bypass [8,40,41]. In order to understand how the glycosylation of the PSI domain affects its trafficking route, we introduced an artificial glycosylation site in cardosin A’s PSI to obtain a structure similar to the one of cardosin B’s and to evaluate if the glycosylation would influence the route taken by PSI-A. The opposite was obtained for PSI-B, where the glycosylation motif was removed and replaced with the amino acid present in cardosin A’s PSI sequence. In contrast to the non-glycosylated PSI-A, which accumulates in the vacuole despite the ER-to-Golgi blockage, the mutated version was retained in the early compartments when co-expressed with SarI^H74L^. Furthermore, the non-glycosylated form of PSI-B was no longer sensitive to the ER-to-Golgi blockage, being able to accumulate in the vacuole even when co-expressed with SarI^H74L^. These results show the relevance of N-glycosylation in protein trafficking. It has been previously proposed that N-glycosylation is not a determinant for the correct trafficking of vacuolar proteins, leading instead to a delay in the trafficking [42,43]. Here, we show that the glycosylation may indeed be important for defining the route taken by proteins, in particular in what concerns their passage through the Golgi. This is true for cardosin A’s PSI, as the difference in sensitivity to ER-to-Golgi blockage of the non-glycosylated and glycosylated form is striking. In the case of PSI-B, however, the results obtained in this report are not as clear cut and a more complex interpretation may be needed. The non-glycosylated form of the protein, despite being able to accumulate in the vacuole, is also detected in the ER. In fact, the effect of the co-expression with SarI dominant negative mutant in the vacuolar accumulation of PSI-B::mCherry is not so different for the glycosylated and non-glycosylated versions. In addition, we have also shown that PSI-B migrates in the Western blot at a higher molecular weight than the PSI-A, and this higher molecular weight is maintained after endo-H digestion. All these observations clearly indicate that there must be another post-translational modification interfering with the sorting and trafficking of PSI-B that must be explored.

### 3.3. Soy PSIs Case–Another Piece of Evidence

As previously commented in this report, the number of aspartic proteinases containing a PSI is lower than the typical or nucellin-like aspartic proteinases [20,21]. More interesting is the fact that most of the PSIs have a glycosylation motif, like cardosin B’s PSI, and only a few are not glycosylated. Terauchi and co-workers (2004, 2006) [28,39] isolated and characterized two aspartic proteinases from soybean, soyAP1 and soyAP2, and also discuss the role of the PSI in the vacuolar targeting of these enzymes. The analysis of the soy PSIs sequence made it apparent that they share the same dichotomy as cardosins’ PSIs: One is glycosylated and the other is not. As a proof of concept, we isolated the soy PSIs and ran the same experiments as for cardosins’ PSIs to check if their trafficking would be affected. Subcellular localization of soy PSI::mCherry chimeric proteins shows accumulation in the vacuole, establishing that both soy PSI domains are VSDs and contain all the necessary information for vacuolar sorting. Furthermore, and like cardosins’ PSIs, soy PSIs show differential sensitivity to ER-to-Golgi blockage when co-expressed with SarI^H74L^, in a glycosylation-dependent manner. Although it was suggested in the report by Terauchi and co-workers [28] that soyAP1′s PSI was not involved in vacuolar-sorting events, the data obtained in this study clearly shows this is not the case. Probably, in the context of the enzyme, the soy PSI is in a conformation that does not allow the vacuolar accumulation of the protein, or it might be subjected to the type of hierarchical regulation that was previously described for cardosin A’s VSDs [9]. It is also possible that the importance of the PSI in the vacuolar sorting information depends on the plant developmental stage, or the organ/cell type where it is being expressed. In fact, the authors show that the removal of the PSI domains altered the APs targeting to the lytic vacuole, but not to the protein storage vacuole. The PSI may, therefore, act as vacuolar signal only in specific conditions or developmental stages. Taken together, the results obtained with the cardosin and soy PSIs allow us to assume that glycosylation may play an important role in the sorting and trafficking of proteins, or at least influence the way proteins leave the ER.

### 3.4. Cardosin B PSI: One VSD, Multiple Pathways?

In former studies, Pereira and colleagues pointed out that the occurrence of multiple VSDs in cardosins (and atypical APs in general) could be related to regulatory mechanisms employed by plants in order to increase these proteases’ functional diversity in different types of cells, tissues and/or developmental stages [5,9]. Our results with PSI B-mediated sorting may come as further confirmation of this hypothesis, as we have observed a differential behavior between vacuolar sorting mediated by this domain on what could be described as a case-by-case mechanism. Isolated PSI-B fused to the mCherry fluorescent reporter was efficiently directed towards the vacuole in a RabF2b-dependent pathway, similar to what was observed with intact cardosins A and B [9,17]. Integration of the PSI-B domain in an entire cardosin structure, independent of which cardosin was tested, however, resulted in a different dynamic behavior as vacuolar sorting shifted from RabF2b-dependent to RabF2b-independent for PSI B-mediated vacuolar sorting. This observation is somewhat surprising, as it seems to imply that the vacuolar trafficking pathway followed by this domain is somehow determined by its overall three-dimensional structure, without compromising its VSD status or efficiency. This result came as further confirmation that plant APs possess multiple VSDs capable of vacuolar sorting through different routes, and that despite similar three-dimensional structures, different PSI domains possess different specificities in terms of protein sorting to the vacuole. Given the importance of protein folding and three-dimensional structure in PSI-mediated functionality [23], it would be feasible to assume that isolation of this domain could give different results. This observation seems to imply that structural changes occurring in this domain could be modulating the route followed towards the lytic vacuole at the TGN-PVC level.

### 3.5. PSI-Mediated Sorting: An Unconventional Vacuolar Sorting Mechanism

In plant cells, the conventional route to the vacuole involves the ER, Golgi and the prevacuolar compartment [44]. However, in the last few years, a more unconventional sorting route to the vacuole that bypasses the Golgi, has been described, which is the case of some vacuolar proteins such as Chitinase A [8] and several membrane proteins [45,46]. However, the mechanisms underneath this Golgi-bypass are still unclear and have only recently started gaining attention [11,47]. The Golgi-bypass is also the case for the route mediated by cardosin A’s PSI. This domain is both sufficient and necessary to direct a secreted protein to the vacuole in an ER-to-Vacuole direct pathway. The mechanisms allowing Golgi bypass are currently unknown, but it is interesting to notice that it is not true for all PSIs. After resolving prophytepsin’s crystallographic structure in 1999, Kervinen and coworkers identified a positively charged ring formed by residues from the PSI domain (Lys320, Lys400 and Arg415; preprophytepsin numbering) and the proteinase’s light chain, that could correspond to a putative receptor-binding site [48]. These residues are not all conserved between PSIs and we could hypothesize that the differential binding of PSIs with vacuolar receptors or ER-resident proteins could be the base of a Golgi-bypass mechanism. This hypothesis is also compatible with the idea that different PSI domains could have different sorting functions due to their glycosylation status, as the highly conserved N-linked glycan is inserted at the periphery of this positively charged ring (Asn399) and could be expected to either interfere or modulate protein–protein interactions taking place at this site. Our current data do not permit so far to determine the features needed for a PSI domain to act or not as VSDs nor to determine which route it will take. Considering that proteins delivered directly to the vacuole would have to be recognized at the ER level, testing the PSI interaction with several ER proteins and even searching for the existence of a PSI receptor, could be a clue to this question. In fact, the results from the BFA treatment presented here are an indication that alterations in ER/Golgi morphology or the re-localization of Golgi and ER-export sites’ components (in this case induced by the drug) affect the vacuolar sorting mediated by PSI-A. In the presence of BFA, PSI-A::mCherry fusion is mostly found in BFA-bodies not being able to accumulate in the vacuole as efficiently as before. It is a clear indication that the PSI-mediated Golgi-bypass is a process initiated and reliant on the ER. It would be exciting to explore in more detail the PSI-A-mediated trafficking in order to gather more data and unveil the general mechanism behind this unconventional pathway.

## 4. Materials and Methods

### 4.1. Plasmids and Vectors

All constructs used in this study were generated by Polymerase Chain Reaction (PCR) using specific primers (Table 1) and a proofreading DNA polymerase (*Pfu* DNA Polymerase, Thermo Scientific). Unmodified Cardosin A [16] and cardosin B [17] were used as templates for all PCR reactions for cardosins’-based cloning and Soybean AP 1 and 2 [39] as a template for PSIs 1 and 2. PCR fragments were initially cloned using the Zero Blunt^®^ cloning kit (Invitrogen) and analyzed by restriction mapping. Positive clones were selected for sequencing using universal M13 primers (Eurofins MWG Operon). Cardosins’ PSIs and mutated versions were then inserted into the *Xba*I and *Sal*I (cardosin A/B-based constructs) sites of the binary vector pVKH18-En6::mCherry [49] and Soybean PSIs into *Xba*I and *Sac*I sites of pMDC83 [50] for expression in plant cells. All the gene constructs were under the action of the cauliflower mosaic virus (CaMV) 35S promoter and NOS terminator. Specific details for each set of constructs are given below.

*Cardosins’ A and B PSIs:* Constructs encoding for PSI-A and PSI-B tagged with mCherry at the C-terminal were already available [9] and were used as a template to generate the glycosylation mutants. Modified primers were designed (Table 1) and mutated forms were obtained using the site-directed mutagenesis technique with *Pfu* DNA Polymerase (Thermo Scientific) coupled to template digestion with *Dpn*I (Fermentas).

*Soy AP’s 1 and 2 PSIs:* SoyAP1 and SoyAP2 cDNA cloned into pBLUESCRIPT II SK were kindly provided by Terauchi and co-workers [39]. In order to obtain the isolated domains, each PSI was amplified by PCR using a specific set of primers that introduced *Sal*I sites flanking the PSI (Table 1) and allow cloning in frame with mCherry into the pMDC83 vector containing SP::mCherry (already available).

*Cardosins’ swapped PSI domains*: To swap the PSI domains between cardosins A and B, they were PCR-amplified (Table 1) in order to delete their native PSI domains, CardosinΔPSI::mCherry, and the PSIs were also isolated by PCR. CardosinΔPSI::mCherry sequences were then ligated to the PSI fragments for obtaining the swapped PSI versions (CdAPSI-B and CdBPSI-A constructs). Additionally, reverse primers were designed in order to delete the cDNA sequences encoding the proteins’ c-terminal peptides and allow in-frame cloning with mCherry-Cardosin A::PSI-B::mCherry∆c-ter and Cardosin B::PSI-A::mCherry∆c-ter. The same primer pairs were used to remove the c-terminal sequence from the un-modified cardosin constructs, Cardosin A::mCherry∆c-ter and Cardosin B::mCherry∆c-ter.

### 4.2. Plant Material

The analysis of the constructs produced was done using the method of *Agrobacterium tumefaciens*-mediated transient transformation of *Nicotiana tabacum* leaf epidermis by infiltration. Seeds of *N. tabacum* cv. SRI Petit Havana were germinated in petri dishes on filter paper and moistened with water. After germination, seedlings were transferred to individual pots with fertilized substrate (SiroPlant) and maintained in a growth chamber with a photoperiod of 16 h light, 60% humidity and 21 °C.

### 4.3. Transient Expression in N. Tabacum Leaves

*Agrobacterium tumefaciens* GV3101::pMP90 was transformed by electroporation, screened by restriction mapping, and used for infiltration of *Nicotiana tabacum* L. cv. Petit Havana SR1, as described by Batoko et al. [36] with the following modification: YEB medium was replaced with LB broth supplemented with 50 µg.mL^−1^ kanamycin. For co-expression experiments, the bacteria harboring the different constructs were mixed prior to infiltration, with the titre adjusted to the required OD600. The used OD600 were as follows: 0.3 for the PSIs and cardosins’ constructs and 0.15 for ST::GFP, GFP::HDEL, SarI^H74L^::YFP, and RabF2b^S24N^.

### 4.4. Protoplasts Isolation From N. Tabacum Leaves

The lower epidermis of 24-h infiltrated leaves was mechanically removed using sharp forceps and incubated in 90-mm petri dishes containing a mixture of 1% (*w/v*) cellulase (Onozuka R10) and 0.25% (*w/v*) macerozyme (Onozuka R10), dissolved in TEX medium [3 mM NH_4_NO_3_, 5 mM CaCl_2_·2H_2_O, 2.4 mM MES and 0.4 M sucrose in Gamborg B5 medium (Duchefa)] at RT, in the dark, for 16 h. The fraction of intact protoplast (floating) was obtained after centrifugation (100× *g*, for 10 min at RT). The fraction was transferred to a 15-mL tube containing two volumes of mannitol/W5 [0.4 M D(-)-mannitol, 1 mM D(+)-glucose, 30.8 mM NaCl, 25 mM CaCl_2_, 1 mM KCl, 0.3 mM MES; pH 5.6–5.8] and gently mixed. Protoplasts were pelleted at 100× *g* for 5 min at RT and the supernatant was carefully removed. The protoplast pellet was resuspended in TEX medium and incubated for the time of the experiment in 6-well plates in the dark, at 21 °C.

### 4.5. Drug Treatments

BFA treatment was performed as described in Soares da Costa et al. 2010 [17]. Briefly, at 36–40 h after leaf infiltration with the desired construct, infected areas of the leaves were infiltrated with 50 µM BFA, prepared in water (stock solution in DMSO). As control, a solution with the dilution of DMSO was used. The infiltrated areas were removed and left to float on the same solution for 2 h, at 21 °C in the dark. For BFA treatment in protoplasts, the volume correspondent to 20 µM of BFA was added to the medium containing protoplasts and incubated in the dark for 2 h, at 21 °C. In the control experiment, the same volume of DMSO was added to the protoplast solution and incubated in the same conditions.

### 4.6. Protein Sample Extraction and Endoglycosidase Assays

Protein extraction from leaf tissue was performed in the presence of two volumes of extraction buffer [50 mM sodium citrate, pH 5.5; 5% SDS (*w/v*); 0.01% BSA (*w/v*); 150 mM NaCl; 2% (*v/v*) β-mercaptoethanol and 10 µL of protease inhibitor cocktail (Sigma-Aldrich)] per 300 mg of tissue sample. The tissue was mechanically disrupted and boiled for 10 min. The samples were then centrifuged at maximum speed for 30 min at 4 °C and the supernatant collected. For Endo-H assays (Endo-Hf, New England Biolabs), 5 µL of total protein extract (approximately 10 µg) was used in each reaction and the protocol provided by the supplier was followed.

### 4.7. Western Blot

SDS-PAGE was performed using a 12% polyacrylamide gel. Five microliters of total protein extract (approximately 10 µg) was loaded on the gel, and 5 µL of CLEARLY Stained Protein Ladder (Takara Bio) was used as a protein molecular weight marker. After electrophoresis, the proteins were transferred to a nitrocellulose membrane with a Tris-glycine-methanol buffer. The membrane was blocked for 1 h in Tris buffered saline supplemented with 5% (*w/v*) skim milk, 1% (*w/v*) bovine serum albumin and 0.6% (*v/v*) Tween 20. A monoclonal antibody against mCherry (Milipore) was used at a 1:1000 dilution to probe the membrane at 4 °C, overnight. Alkaline phosphatase conjugated secondary antibody (Vector) was used at a 1:1000 dilution for 1 h at room temperature, and the proteins exposed with Novex AP Chromogenic substrate (Invitrogen), according to the manufacturer’s protocol.

### 4.8. Confocal Laser Scanning Microscopy-Image Acquisition and Analysis

Images were acquired with an inverted SP2 (for single image acquisition) or SP5 (for co-localizations) Leica laser scanning microscope. Pieces of leaf were sampled from the infiltrated area in a random fashion and mounted in water. The 561 nm laser line was used for excitation of mCherry, whereas the 488 line was used for exciting GFP and YFP. Images were processed using ImageJ/Fiji software.

## 5. Conclusions

In this report, we show that a post-translational modification like N-linked glycosylation can interfere with the sorting and trafficking of proteins to the vacuole, as the non-glycosylated PSI-A is able to bypass the Golgi and accumulate directly in the vacuole, while (glycosylated) PSI-B is not. This work also confirmed the role of the PSI in protein sorting and a deeper study into its function as an unconventional vacuolar determinant will definitely shed some light on the understanding of the mechanisms behind unconventional protein sorting to the vacuole. Moreover, considering previous results and the data presented here, it is plausible to suggest that the PSIs’ role in vacuolar targeting can be dependent on the specific function of the proteinase it is integrating and on the developmental stage and metabolic activity of the tissue it is expressed in.

## Figures and Tables

**Figure 1 plants-08-00312-f001:**
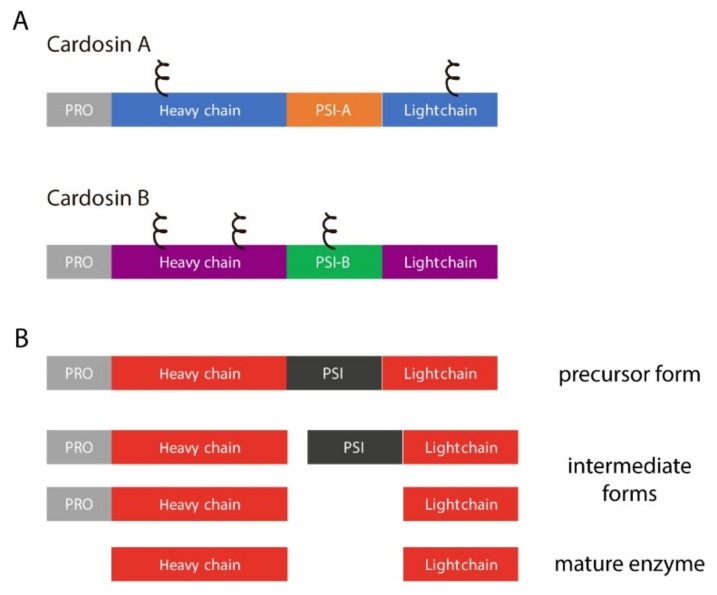
Cardosins’ organization and processing steps. (**A**) schematic representation of cardosins A and B, highlighting the different domains and predicted glycosylation sites (ξ); PSI–Plant Specific Domain; PRO–prosegment; (**B**) Sequential processing steps usually taken by aspartic proteinases in order to acquire the mature form.

**Figure 2 plants-08-00312-f002:**
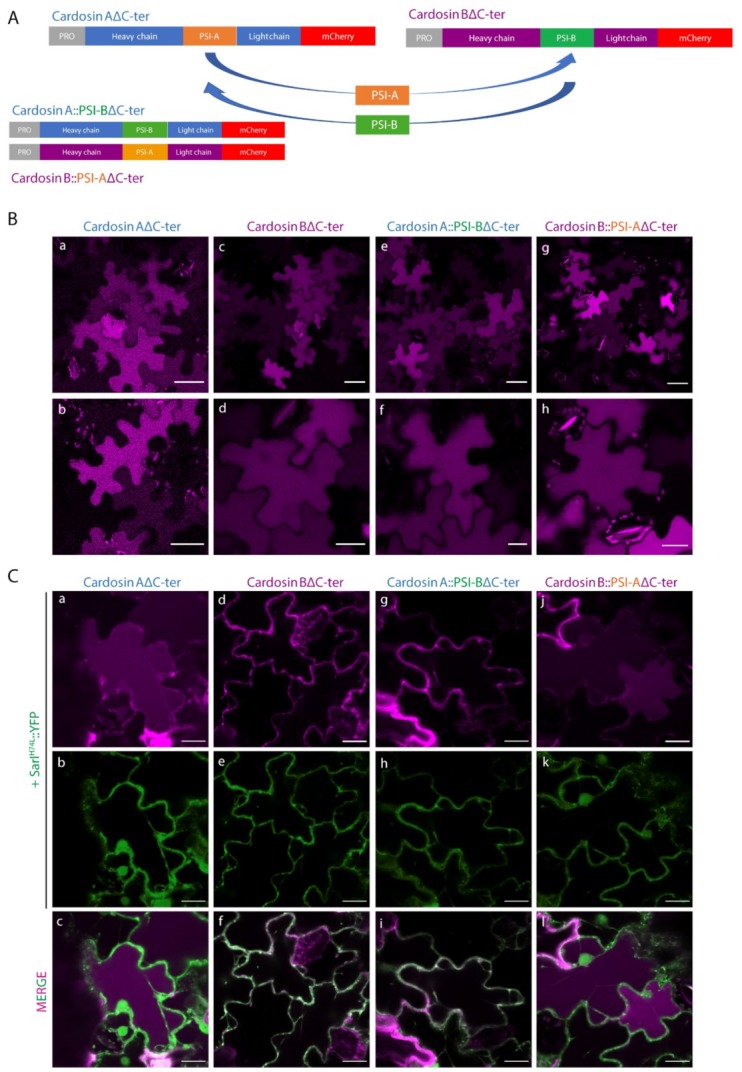
Subcellular localization of cardosins A and B and their PSI-swapped versions in *N. tabacum* epidermal cells. (**A**) Schematic representation of the fusion proteins used in this assay. (**B**) Expression in tobacco epidermal cells evidencing the vacuolar localization of all fusion proteins: a, b–Cardosin A∆C-ter; c, d–Cardosin B∆C-ter; e, f–CardosinA::PSIB∆C-ter; g, h-CardosinB::PSIA∆C-ter. (**C**) Co-expression of the dominant negative mutant SarI^H74L^::YFP with: a–c-Cardosin A∆C-ter; d–f-Cardosin B∆C-ter; g–i-CardosinA::PSIB∆C-ter; j–l-CardosinB::PSIA∆C-ter. Note the differences in vacuolar accumulation depending on the PSI domain present. All observations and images were acquired 3 days post-infiltration. Images were analyzed and processed using ImageJ/Fiji software. Scale bars: (**B**)—upper panels, 50 µm and lower panels 20 µm; (**C**)—20 µm.

**Figure 3 plants-08-00312-f003:**
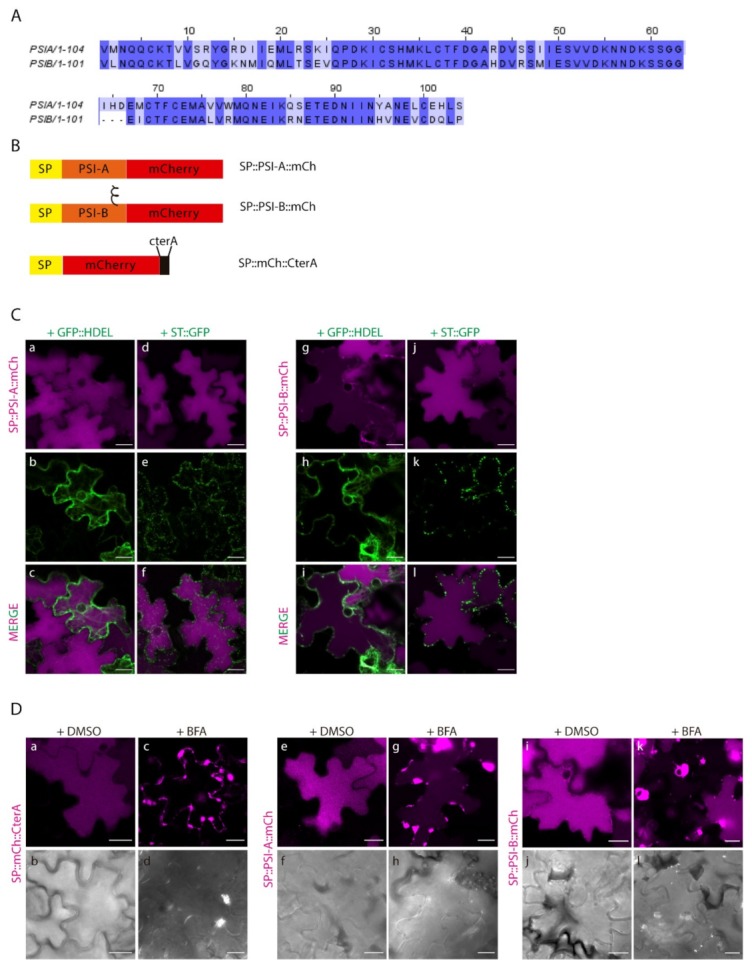
Expression of PSI-A and PSI-B in *N. tabacum* epidermal cells reveals subtle differences between the two. (**A**) Alignment of cardosins’ A and B PSIs, showing the similarity between them. (**B**) Schematic representation of the fusion proteins designed for this study. (**C**) Co-expression of SP::PSI-A::mCh with the ER marker GFP::HDEL (a–c) and the Golgi marker ST::GFP (d–f) and of SP::PSI-B::mCh with GFP::HDEL (g–i) and ST::GFP (j–l). Single images for both channels and the composite image are provided. (**D**) Expression of PSI-A (e–h) and PSI-B (i–l) followed by Brefeldin A treatment. SP..mCh::CterA (a–d) was used as a positive control, and DMSO was used as a negative control for the experiment. DIC images of the same cells were also acquired to evaluate the cell morphology. All observations and images were acquired 3 days post-infiltration. Images were analyzed and processed using ImageJ/Fiji software. Scale bars: (**C**) and (**D**)—20µm.

**Figure 4 plants-08-00312-f004:**
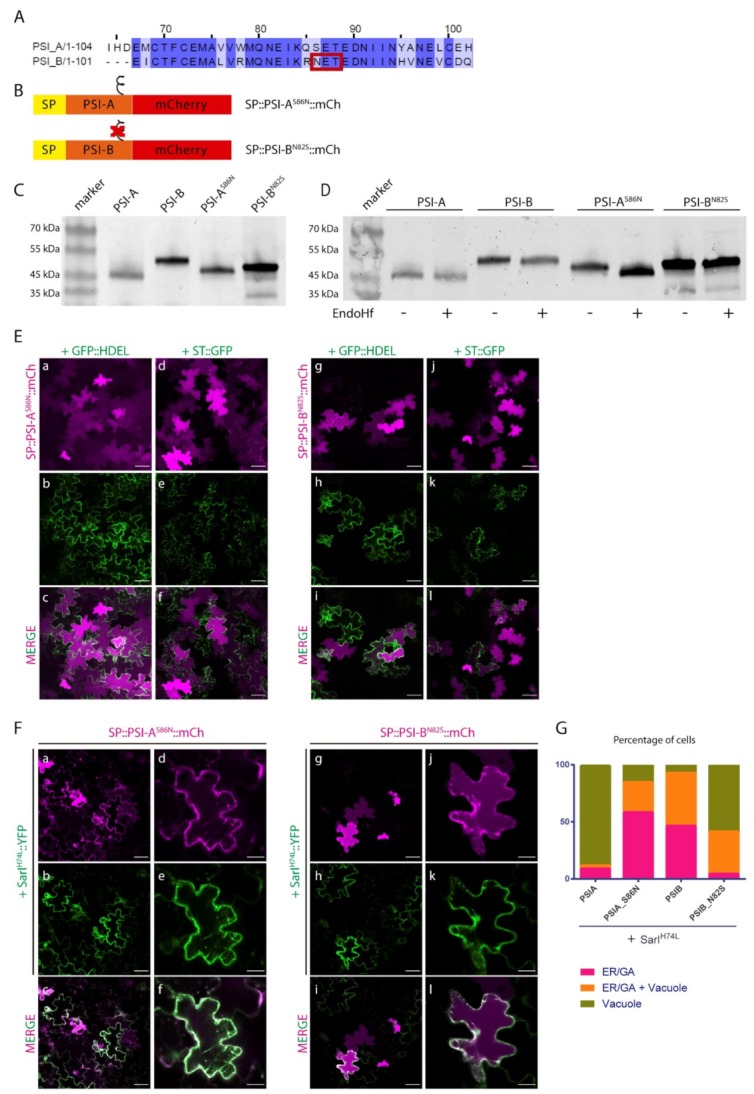
The expression of PSIs’ glycosylation mutants in *N. tabacum* epidermal cells highlights a putative role for N-glycosylation in trafficking. (**A**) Alignment of cardosins A and B PSIs, showing the glycosylation motif on PSI-B–red box. (**B**) Schematic representation of the mutated versions produced. The symbol ξ represents the glycosylation motif. (**C**) Western blot of the native and mutated versions of the PSIs, evidencing the differences in migration between them. (**D**) Endoglycosidase H assay result of all PSIs. Note the shift in protein migration for PSI-A^S86N^. (**E**) Co-expression of SP::PSI-A^S86N^::mCh with the ER marker GFP::HDEL (a–c) and the Golgi marker ST::GFP (d–f) and SP::PSI-B^N82S^::mCh with GFP::HDEL (g–i) and ST::GFP (j–l). Single images for both channels and the composite image are provided. (**F**) Co-expression of SP::PSI-A^S86N^::mCh (a–f) and SP::PSI-B^N82S^::mCh (g–l) with the dominant negative mutant SarI^H74L^::YFP indicate differences in the sensitivity of the mutants to this blockage. (**G**) Bar graph comparing the localization between the mutated and native PSI forms under the effect of SarI^H74L^::YFP. For the quantification, we considered that the total number of cells with fluorescent signals defines a 100% value, and among this population, the different localization patterns were then scored. All observations and images were acquired 3 days post-infiltration. Images were analyzed and processed using ImageJ/Fiji software. Scale bars: (**D**)—50 µm; (**E**)—left panels, 50 µm, and right panels, 20 µm.

**Figure 5 plants-08-00312-f005:**
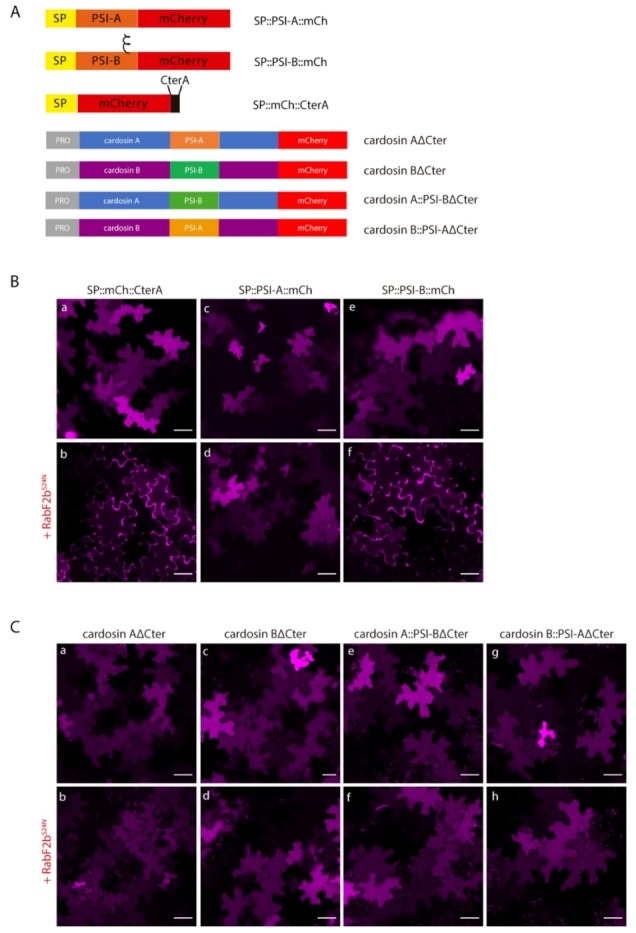
Co-expression of isolated PSIs and cardosin constructs with the dominant negative mutant of RabF2b. (**A**) Schematic representation of the constructs used in this assay. (**B**) Co-expression of SP::PSI-A::mCh (c–d) and SP::PSI-B::mCh (e–f) with the dominant negative mutant RabF2b^S24N^. SP::mCh::C-terA (a–b) was used as a positive control for the efficiency of the mutant. Note the accumulation in the cell wall in the positive control and SP::PSI-B::mCh, but not for SP::PSI-A::mCh. (**C**) Co-expression of RabF2b^S24N^ with cardosin fusion proteins: (a–b) Cardosin A∆Cter, (c–d) Cardosin B∆Cter, (e–f) Cardosin A::PSIB∆Cter, and (g–h) Cardosin B::PSIA∆Cter. All the proteins are insensitive to the blockage of this specific pathway. All observations and images were acquired 3 days post-infiltration. Images were analyzed and processed using ImageJ/Fiji software. Scale bars: 50 µm.

**Figure 6 plants-08-00312-f006:**
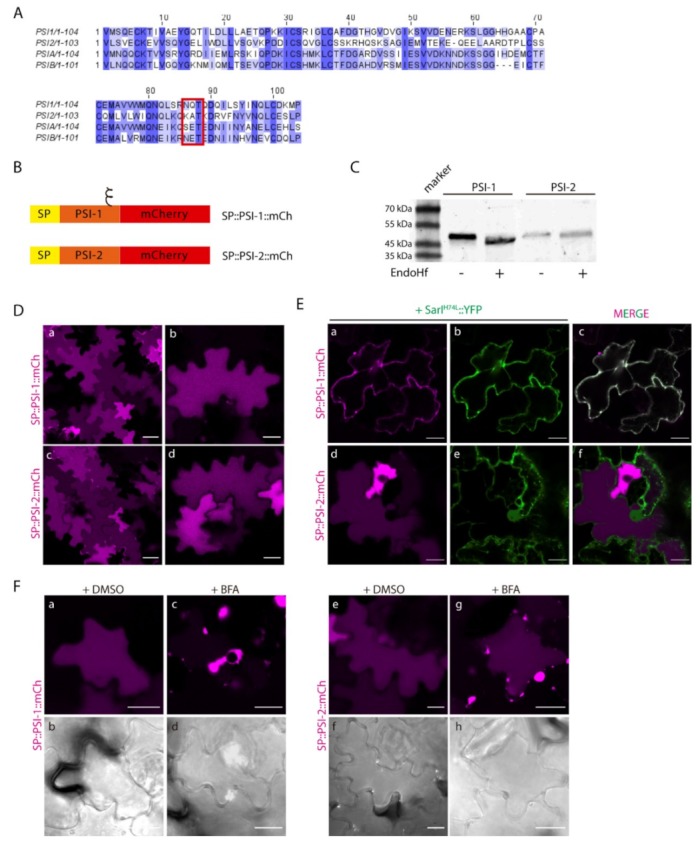
Expression of Soybean PSIs in *N. tabacum* epidermal cells presents the same dichotomy as cardosins’. (**A**) Alignment of Soybean (PSI-1 and PSI-2) and cardosin (PSI-A and PSI-B) PSIs. The sequence conservation is not high, but PSI-1 is predicted to be glycosylated as PSI-B–red box. (**B**) Schematic representation of the two constructs designed for this experiment. (**C**) Western blot obtained after an endoglycosidase H assay for PSI-1 and PSI-2, where a shift-down in the band corresponding to PSI-1 is evident. (**D**) Vacuolar localization of both constructs in tobacco epidermal cells: (a–b) SP::PSI-1::mCh, (c–d) SP::PSI-2::mCh. (**E**) Co-expression of SP::PSI-1::mCh (a–c) and SP::PSI-2::mCh (d–f) with the dominant negative mutant SarI^H74L^::YFP shows differences in the sensitivity of the mutants to this blockage. (**F**) Expression of SP::PSI-1::mCh (c–d) and SP::PSI-2::mCh (g–h) in tobacco cells followed by Brefeldin A treatment. DMSO was used as a negative control (a–b, e–f). DIC images of the same cells were also acquired to evaluate cell morphology. All observations and images were acquired 3 days post-infiltration. Images were analyzed and processed using ImageJ/Fiji software. Scale bars: (**D**)—left panels, 50 µm, right panels, 20 µm; (**E**)—20 µm; (**F**)—20 µm.

**Table 1 plants-08-00312-t001:** Oligonucleotides and template DNA used to produce the constructs used along this project.

Construct to Make	Oligonucleotide Fwd	Oligonucleotide Rev	Template DNA
SP::PSI-A^S86N^::mCherry	GGATGCAAAACGOLGIAATCAAACAAAACGAGACTGAAGATAAC	GTTATCTTCAGTCTCGTTTTGTTTGATTTCGTTTTGCATCC	SP::PSI-A::mCherry
SP::PSI-B^N92S^::mCherry	GCAGAATGAAATCAAACGAAGCGAGACTGAAGATAACATAA	TTATGTTATCTTCAGTCTCGCTTCGTTTGATTTCATTCTGC	SP::PSI-B::mCherry
SP::PSI-1::mCherry	ACGTCGACTGTTATGAGCCAAGAATGCAAGACC	TTGTCGACGCACCACCTGCAGCACCACCGCTAGGCATTTTATCGC	SoyAP1
SP::PSI-2::mCherry	ACGTCGACTGTTCTCAGTGTCGAATGTAAGGAAGTC	TTGTCGACGCACCACCACTTGGCAGGCTCTCAC	SoyAP2
PSI-A (for cardosin A::PSI-B::mCherry)	GTCATGAACCAGCAATGCAA	GGATAAGTGTTCACACAACTC	Cardosin A
PSI-B (for cardosin B::PSI-A::mCherry)	TTAAACCAACAATGCAAAACATTGG	TTCTGCACTTGAAGTGGGTA	Cardosin B
Cardosin A∆PSI::mCherry	ACTTCATCTGAAGAATTACAAG	CCCGTTAGCGCCAATTGCATGATT	Cardosin A
Cardosin B∆PSI::mCherry	TCGATAGTAGACTGCAATGG	AACCCCTTTTGCACCAATTG	Cardosin B
Cardosin A::mCherry∆c-ter	TCTAGAGCCGCCACCATGGGTACCT	GTCGACGCTAGTAAATTGCCATAATCAAACACTGTG	Cardosin A
Cardosin A::PSI-B::mCherry∆c-ter	TCTAGAGCCGCCACCATGGGTACCT	GTCGACGCTAGTAAATTGCCATAATCAAACACTGTG	Cardosin A::PSI-B
Cardosin B::mCherry∆c-ter	CATCTAGACTCGAGCCACCATGGGAACCCCAATCAAAGCAAACG	ACGTCGACTTTAACTTGCCATAATCG	Cardosin B
Cardosin B::PSI-A::mCherry∆c-ter	CATCTAGACTCGAGCCACCATGGGAACCCCAATCAAAGCAAACG	ACGTCGACTTTAACTTGCCATAATCG	Cardosin B::PSI-A

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
