# Peer review of "N-Linked Glycosylation Modulates Golgi-Independent Vacuolar Sorting Mediated by the Plant Specific Insert"

_plants, 2019, doi:10.3390/plants8090312_

Round 1
Reviewer 1 Report
The manuscript by Vieira et al. is based on previous articles published by this research group on the Plant Specific insert (PSI) domain of the cardosin aspartic proteinases and its role as a vacuolar sorting signal. Recently, they have demonstrated that cardosin A has two vacuolar sorting determinants, the C terminal VGFAEAA peptide (the dominant signal) and the PSI, which mediate two different trafficking pathways to the vacuole, the conventional secretory pathway and an unconventional route, respectively (Pereira et al. 2013).
In this manuscript they want to investigate the differences between cardosin A and B PSIs in terms of post-translational modifications, and the consequences of these modifications on protein trafficking, using a transient expression system represented by Nicotiana tabacum leaves infected with Agrobacterium tumefaciens.
The topic of the manuscript is very interesting, but the experimental part, in particular that on glycosylation, needs to be revised because some controls are missing, and the results obtained with BFA treatment and Endo-H digestion are contradictory.
Here is a list of my suggestions to improve the manuscript:
The English language is good, still a minor spell check is required. For example, at line 59 the sentence “reports have been made where this domain..” could be changed in “reports indicated that this domain..” or at line 129 where the word “since” is repeated twice and the second time could be changed in “as”. At line 134, the sentence “To add more data to this discussion” should be revised.
The part comprised between lines 297 and 312 should be moved in the Introduction section, at line 43, to introduce cardosins and PSI. Moreover, the part between lines 43 and 49 should be get together with lines 60 – 66 because here it is introduced the same concept, i.e. the characterization of PSI as vacuolar signal. At line 38, I suggest adding the comment that vacuolar delivery of vacuolar proteins can be also promoted very early in the sorting process, when polypeptides are still contained within the ER (e.g. The Plant Journal 2010, 61 (5), 782-791).
The Figures must be improved.
Each protein fusion should be indicated in the figures with a name and this name must be used in the text to avoid confusion and facilitate the reader. For example, in Figure 2, the protein fusion SP-PSI-AS86N-mCherry is reported as SP::PSI-AS86N::mCh and at line 160 as glycosylated PSI-A::mCherry. In Fig. 4 the C terminal PSI-A peptide is indicated as c-ter or C-terA.
All the Confocal Laser Scanning images should be colored, and each panel should have a letter to identify it in order to use the symbol in the text (e.g. Figure 1C, panels a-f, with coexpression of SP-PSI-A-mCherry and GFP-HDEL in Fig. 1C, panels a-c).
All the cartoons referred to the subcellular compartments in the figures should be in black and white. The symbol of glycosylation in the schematic drawing of the protein fusions should be described in the figure legend. In the legends the word “constructions” (line 99) should be replaced with “ fluorescent chimaeras” or “fusion proteins”.
Replace at line 87 “Figure 2B” with “Figure 1B”. Indicate in Fig. 2B-C the molecular weight in kDa of the proteins. Figure 2F, reported at line 174, is not present in Figure 2, therefore add the symbol “F” to the figure.
In Fig. 3B-C the fusion proteins should be indicated with a name, not with the drawings which are difficult to see (too small).
In Fig. 4A the style of the letters (size and color) in the fusion proteins should be level out.
Section 2.1 should be revised because the data of Fig. 1D have been already described in the article Pereira et al. 2013 (Figures 4 and 7), thus Fig. 1D could be transformed in a supplementary figure. At line 93 the word “vacuole” should be replaced with “Golgi”. The control SP-mCherry-C-terA should be clearly described in the text, writing what is expected from it. In addition, this control must be added to Figure 1E, which is highly controversial because, as demonstrated in several reports (e.g. Batoko etr al. 2000), BFA treatment is used, as well as the dominant-negative RabD2aN121I mutant, to investigate if a protein is transported through the Golgi. From Fig. 1E, the reader understands that BFA treatment blocks the transport of both SP-PSI-A-mCherry and SP-PSI-B-mCherry, which is different from the results of Fig. 1D. For this reason, the results on coexpression of constructs of Fig. 1D with RabD2aN121I mutant should be shown in another supplementary figure. Moreover, in order to clarify the results of Fig. 1E, the BFA treatment should be performed using BFA at 5 -10 μg/mL (instead of 14 μg/mL or 50 μM) and, in addition to SP-mCherry-C-terA, SP-PSI-A-mCherry and SP-PSI-B-mCherry, with the following constructs to understand the influence of the native protein structure: CdAdeltaC-ter-mCherry (Pereira et al. 2013) and CdBdeltaC-ter-mCherry. The part comprised between lines 107 and 111 should be moved at the beginning of section 2.2 because it concerns the glycosylation.
The same observations about the BFA treatment can be made for the soybean PSI fusion proteins (line 281).
Section 2.2. At line 137, the sentence “The output in a Western blot, would be a shift in molecular weight upon digestion with EndoH” should be replaced with “The output in a Western blot would be a shift in molecular weight of proteins not transported through the Golgi upon digestion with EndoH”. At line 141, the sentence “consistent with the presence of high-mannose glycans acquired in the ER” is not correct. Both glycosylated PSI-A and PSI-B acquired in the ER high-mannose glycans, but since the enzymes responsible for glycoprotein acquisition of complex glycan structures are located in the Golgi, Fig 2C strongly suggests that the traffic to the vacuole of the glycosylated PSI-A fusion protein does not depend on Golgi-mediated delivery, whereas traffic to the vacuole of PSI-B fusion protein depends on Golgi-mediated delivery. This is apparently in contrast with the results of Fig. 2E, which indicate that SarIH74L dominant negative mutant blocks the ER-to-Golgi trafficking of glycosylated PSI-A. To better understand the role of glycosylation in the transport of PSIs fusion proteins I suggest repeating the experiments of Fig. 2 with new fusion proteins where mutated PSI are inserted into the native protein, for example: CdA-PSI-AS86N-deltaC-ter-mCherry and CdB-PSI-BN82S-deltaC-ter-mCherry.
Section 2.3 should be moved at the beginning of the Results section ant transformed in section 2.1. With this reorganization, the data of Fig. 1 A-C should be inserted after the results of line 189, while the data of Fig. 1 D-E should be inserted after the results of line 200. The results described at line 210 for RabD2aN121I must be shown in a supplementary figure.
At line 241, section 2.4, the sentence “These observations contrast with the ones obtained for the isolated PSI-B domain..” suggests that in the manuscript the authors must always include in the experiments gene constructs with both the isolated domains and the domains in the context of the native protein.
In the Discussion, the title of section 3.2 is “The significance of glycosylation in sorting”. However, in the manuscript the results on the role of glycosylation in sorting are not clear. Additional experiments are necessary, as I suggested above, to clarify the role of glycosylation because this part represented the novelty of the manuscript.
Materials and Methods. In section 4.1, the authors should describe which are the promoters of the different gene constructs utilized for the transient transformation. At line 507, please described the composition of the solution for the BFA treatment. At line 519, how many μg of proteins are loaded in the gel?
At line 540 of the Conclusions, the sentence “N-linked glycosylation can interfere with the sorting and trafficking of vacuolar sorting determinants to the vacuole as the non-glycosylated PSI’s are able to bypass the Golgi and accumulate directly in the vacuole” is not supported by the results and must be reformulated after additional experiments.
Reviewer 2 Report
A manuscript: „The Plant Specific Insert as an unconventional signal in the route to plant vacuoles“ by authors Vanessa Vieira, Bruno Peixoto, Mónica Costa, Susana Pereira, José Pissarra, and Cláudia Pereira brings new and interesting information on the unconventional route of the vacuolar cargo that involves bypassing the Golgi and is dependent on N-glycosylation state of the Plant Specific Insert (PSI) sequences, as demonstrated for the aspartic proteinase cardosin A and B. The manuscript is original, novel, nicely written, and performed experiments nicely assess its main message.
However, several items should be solved prior to the manuscript acceptance:
One is the title – it should be less general since it weakens the relevance of presented data, it should be one sentence description of the results of this work.
Line 68 we cloned the PSIs…
Line 87 Figure 1B
Line 90 – there is no fluorescence overlap of constructs with ER and Golgi marker
Major concern is BFA treatment, it may be irrelevant since the treatments was quite harsh and incomparable to described brief treatments that lead to formation of structurally ordered BFA bodies. Could authors provide in supplementary data bright field images of treated cells under these conditions? If cells are obviously stressed and dying, these data on BFA experiments should be removed from the manuscript.
Line 165 …negative SarI mutant
Line 355 …non-glycosylated PSI-A…
The arrangement and clearness of the figures should be improved:
Figure 1
Figure 1A should contain a scheme of cardosins and their structure (similar to how it is presented for constructs) with stress on PSIs positions and other relevant motives for cardosin function.
Could 1B be better aligned? Depiction of the methodology is unnecessary.
1C -1E and also all other microscopy figures in the manuscript - it would be useful if the two channels, green and red/violet were presented colored. The colored scheme of the pathway in 1C is too trivial.
Figure 2
2A part looks a bit disorderly, could the alignment and constructs schemes be in one line?
2B, C - it is necessary to put kDa/marker information, so that we know how big is the shift, or how much are additional weaker bands smaller.
2D, E - the naive schemes of the pathway are redundant and unnecessary. Data on statistics should be encompassed in the figure legend.
Figure 3
3A scheme is ok, but again non-aligned
3B and C – the schematic presentations of construct above images are too small and letters unreadable, maybe there could be just the names of the constructs, they are not that long and complicated.
Figure 4
4A redundant with 3A, methodology scheme unnecessary; 4B ok.
Figure 5
5C molecular weights?
5E and F schemes redundant and unnecessary.
Round 2
Reviewer 1 Report
I attached a pdf file with a further suggested revision (sentences to modify are highlighted in yellow with an additional comment).

Author Response
Thank you for your comments. All questions were addressed in the new version of the manuscript.